# The voltage-gated potassium channel *Shaker* promotes sleep via thermosensitive GABA transmission

Ji-hyung Kim[1,3], Yoonhee Ki[1,3], Hoyeon Lee[1], Moon Seong Hur[1], Bukyung Baik[1], Jin-Hoe Hur[2], Dougu Nam[1] & Chunghun Lim[1✉]

Genes and neural circuits coordinately regulate animal sleep. However, it remains elusive how these endogenous factors shape sleep upon environmental changes. Here, we demonstrate that *Shaker* (*Sh*)-expressing GABAergic neurons projecting onto dorsal fan-shaped body (dFSB) regulate temperature-adaptive sleep behaviors in *Drosophila*. Loss of *Sh* function suppressed sleep at low temperature whereas light and high temperature cooperatively gated *Sh* effects on sleep. *Sh* depletion in GABAergic neurons partially phenocopied *Sh* mutants. Furthermore, the ionotropic GABA receptor, *Resistant to dieldrin* (*Rdl*), in dFSB neurons acted downstream of *Sh* and antagonized its sleep-promoting effects. In fact, *Rdl* inhibited the intracellular cAMP signaling of constitutively active dopaminergic synapses onto dFSB at low temperature. High temperature silenced GABAergic synapses onto dFSB, thereby potentiating the wake-promoting dopamine transmission. We propose that temperature-dependent switching between these two synaptic transmission modalities may adaptively tune the neural property of dFSB neurons to temperature shifts and reorganize sleep architecture for animal fitness.

[1] School of Life Sciences, Ulsan National Institute of Science and Technology (UNIST), Ulsan 44919, Republic of Korea. [2] UNIST Optical Biomed Imaging Center, UNIST, Ulsan 44919, Republic of Korea. [3] These authors contributed equally: Ji-hyung Kim, Yoonhee Ki. ✉email: clim@unist.ac.kr

Sleep is an essential behavior that is manifested by highly conserved physiological features across animal species. These include behavioral quiescence in a sleep-specific posture, elevated arousal threshold, daily regulation of the sleep-wake cycle by circadian clocks, and sleep homeostasis. Nonetheless, the evolutionary distance between different animal species does not necessarily correlate with similarities or differences in their daily sleep behaviors. It is thus likely that animal sleep has adaptively evolved according to various ecological circumstances in addition to the sleep-regulatory components encoded by each genome.

A *Drosophila* model of sleep has been established to understand how genes, neurotransmitters, and neural activity collectively regulate sleep behaviors[1–3]. A missense mutation, *minisleep* (*mns*), in the gene encoding the evolutionarily conserved voltage-gated potassium channel *Shaker* (*Sh*) has been identified via a forward genetic screen[4]. This mutation is known to cause wakefulness. *Sh*-dependent sleep regulation is further supported by the short sleep phenotypes observed in mutants of two *Sh*-related genes, *Hyperkinetic* (*Hk*) and *sleepless* (*sss*). HK and SSS proteins associate with the SH channel and potentiate its activity or stability[5–9]. Accordingly, the SH complex represents one of the major sleep-promoting pathways in *Drosophila*, although *Sh*-independent sleep regulation by *sss* has also been reported[10]. Another sleep-promoting pathway involves the inhibitory neurotransmitter γ-aminobutyric acid (GABA). Indeed, pharmacological enhancement of GABA transmission has been shown to promote sleep[11,12]. In addition, genetic manipulation of GABA receptors in a subset of *Drosophila* neurons disrupts specific aspects of daily sleep behaviors[13–16] and masks the sleep-promoting effects of a GABA receptor agonist[12].

On the other hand, two monoamine neurotransmitters, dopamine (DA) and octopamine (OA, closely related to nor-epinephrine in mammals), contribute to wake-promoting pathways in *Drosophila*. Transgenic excitation of either dopaminergic or octopaminergic neurons induces insomnia-like behaviors with distinguishable daily sleep profiles[17–19]. A specific subset of sleep-regulatory neurons has been mapped; these neurons express subtypes of DA or OA receptors, respectively, and mediate the wake-promoting effects of their ligands[18–21]. In particular, two groups of wake-promoting dopaminergic neurons, PPL1 and PPM3, have been shown to project onto sleep-promoting dorsal fan-shaped body (dFSB) neurons in the central complex of adult fly brain[18,19]. D1-like DA receptors, *Dop1R1* and *Dop1R2*, transmit the inhibitory dopaminergic input to dFSB neurons and suppress their sleep-promoting neural activity[18,19,21]. Sleep state further modulates sleep-regulatory inputs into dFSB neurons or the excitability of dFSB neurons, suggesting their roles in sleep homeostasis[18,22,23]. Accordingly, dFSB neurons are considered as analogous to sleep-promoting ventrolateral preoptic nucleus (VLPO) in mammalian brains[18,24].

In fact, the opposing effects of the voltage-dependent *Sh* and voltage-independent leak potassium channel *Sandman* set the electrical property of dFSB neurons[21]. *Sh* depletion in dFSB neurons suppresses the generation of repetitive neural activity while negligibly affecting their DA responses. Consistently, dFSB-specific depletion of the SH complex, including HK and SSS, decreases daily sleep amount. By contrast, *Sandman* depletion in dFSB neurons blocks their entry into electrical quiescence upon prolonged DA transmission, thereby inducing sleep. Sleep need is further sensed by the elevation of mitochondrial reactive oxygen species and the subsequent oxidation of nicotinamide adenine dinucleotide phosphate bound to HK[25]. The latter leads to high-frequency spiking in dFSB neurons that induces sleep, underlying their role in sleep homeostasis.

While genetic and neural components of sleep regulation have been defined, the mechanisms by which endogenous sleep-regulatory pathways sense external sleep-modulatory cues and adjust sleep behaviors remain elusive. Temperature is an environmental factor that affects sleep acutely and reversibly. Animals keep a 24-hour periodicity in their circadian rhythms over a physiological range of ambient temperatures. This phenomenon is an important clock property of circadian rhythms and is known as temperature compensation[26]. By contrast, mid-day siestas and nighttime insomnia in hot summer exemplify the temperature-dependent plasticity of sleep in humans. These adaptive changes in sleep-wake cycles are also present in *Drosophila*[27–29]. Both circadian clock-dependent and -independent mechanisms are implicated in shaping the sleep architecture in immediate response to heat[30,31]. Nonetheless, genetic and neural bases underlying the clock-independent control of sleep by temperature or the long-term effects of temperature shifts on sleep behaviors are largely unknown.

## Results

**Sh mutants display temperature-sensitive short sleep.** To understand how ambient temperature affects sleep behaviors in *Drosophila*, we designed a behavioral test with a temperature shift (Fig. 1a). We first measured baseline sleep in individual flies at 21 °C in 12-h light:12-h dark (LD) cycles. We then elevated the ambient temperature to 29 °C and continuously monitored their sleep behaviors. This experimental setup allowed us to trace the temperature-dependent behavioral plasticity in individual flies. In addition, we could distinguish between immediate arousal responses to the temperature shift at midnight (i.e., 4 h after lights-off) and chronic effects of the high temperature on baseline sleep in the following days.

In wild-type flies, the temperature shift from 21 to 29 °C acutely suppressed nighttime sleep (D sleep); however, the daily amount of baseline sleep subsequently became comparable between 21 and 29 °C (Supplementary Fig. 1a, b). More notable effects of temperature were observed on sleep bout numbers and average sleep bout length (ABL). In particular, high temperature increased sleep bout numbers (Supplementary Fig. 1c) while substantially shortening nighttime ABL (Supplementary Fig. 1d). Accordingly, daily sleep architecture was modulated at 29 °C by long daytime sleep (L sleep) and D sleep fragmentation (Supplementary Fig. 1a, b). Additional change included longer latency to D sleep onset at 29 °C (Supplementary Fig. 1e). The temperature-driven plasticity of sleep behaviors was consistent with nocturnal activities in wild-type flies at high temperature[27–29] (Supplementary Fig. 1f). Nonetheless, our assessment of waking activity (i.e., activity counts per minute awake) suggested that ambient temperatures were less likely to have an effect on locomotion per se (Supplementary Fig. 1g).

We next examined whether mutations in sleep-regulatory genes could affect sleep behaviors differentially at either temperature. Indeed, we found that loss of *Sh* function caused short sleep phenotypes in a temperature-dependent manner. Male flies hemizygous for the *mns* mutation (*Sh*[mns]) displayed short and fragmented sleep at 21 °C (Fig. 1b, c and Supplementary Fig. 2), consistent with the previous observations[4]. Intriguingly, *Sh* mutants did not display any L sleep phenotypes at 29 °C. Their short D sleep was also partially rescued by high temperature since we detected a significant interaction between temperature and *Sh* mutation on D sleep amount (Fig. 1c, $P < 0.0001$ by Aligned ranks transformation ANOVA). These results thus indicate that *Sh* promotes sleep more potently at low temperature.

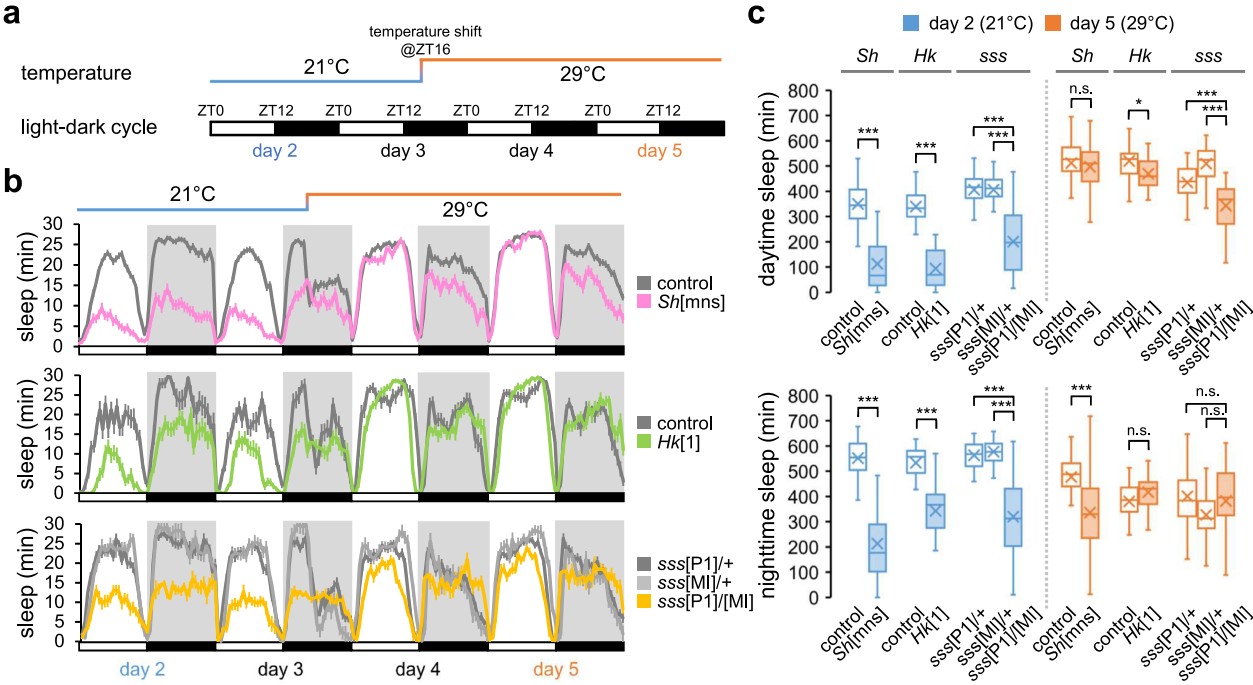

**Fig. 1 _Sh_ and _Hk_ mutants display short sleep phenotypes in a temperature-sensitive manner. a** A schematic of 12-h light (white bars):12-h dark (black bars) cycles and a temperature shift for monitoring temperature-adaptive sleep behaviors. **b** Sleep profiles of _Sh_ (pink line, n = 68), _Hk_ (green line, n = 27), _sss_ mutants (yellow line, n = 29), and their controls (gray lines, n = 20–122). Sleep behaviors were analyzed in individual male flies and averaged for each genotype. Error bars indicate SEM. **c** Box plots represent the total amounts of L or D sleep on day 2 (21 °C, blue boxes) versus day 5 (29 °C, orange boxes) (n = 20–122). Each box plot ranges from lower Q1 to upper Q3 quartile; crosses and horizontal lines inside each box indicate mean and median values, respectively; whiskers extend to minimum or maximum values of 1.5 × interquartile range. n.s., not significant; *P < 0.05, ***P < 0.001 as determined by Aligned ranks transformation ANOVA, Wilcoxon signed-rank test.

We further investigated if sleep mutant phenotypes in _Sh_-relevant genes were also sensitive to ambient temperature. As described previously[8,9], both _Hk_ and _sss_ mutants showed _Sh_-like, short sleep phenotypes at 21 °C (Fig. 1b, c and Supplementary Fig. 2). However, these two sleep mutants behaved very differently at 29 °C. Loss of _Hk_ function largely phenocopied _Sh_ mutation, since short sleep in _Hk_ mutants was significantly rescued at 29 °C (P < 0.0001 for temperature x genotype interaction on either L or D sleep by Aligned ranks transformation ANOVA). By contrast, _sss_ mutants showed short L sleep even at 29 °C and additive effects of temperature and _sss_ mutation were detected on L sleep (P = 0.6722 for _sss_[MI] heterozygous backgrounds by Aligned ranks transformation ANOVA). Given the lack of _Sh_ phenotypes in L sleep at 29 °C, these observations suggest that _sss_ promotes L sleep independently of its regulatory role in _Sh_ function at high temperature. In this regard, transgenic expression of wild-type _sss_ in cholinergic neurons of _sss_ mutants rescued short sleep but not leg-shaking phenotypes relevant to _Sh_[6]. It has been further demonstrated that _sss_ antagonizes nicotinic acetylcholine receptors and suppresses cholinergic transmission in wake-promoting neurons[10].

**Light masks _Sh_ effects on sleep at high temperature.** High temperature suppresses D sleep consolidation, which may lead to the lengthening of L sleep duration as a consequence of sleep homeostasis[32,33]. We reasoned that a similar mechanism could underlie temperature-sensitive L sleep phenotypes in _Sh_ mutants. Accordingly, we compared wild-type and _Sh_ mutant sleep in constant light (LL) or constant dark (DD). LL abolishes circadian rhythms in _Drosophila_[34], thus disrupting daily sleep-wake cycles. Conversely, DD allows circadian gene expression and locomotor

rhythms in the free-running condition while avoiding any direct behavioral responses to light transitions.

_Sh_ mutants displayed short sleep phenotypes in LL at 21 °C (Fig. 2a, b), indicating that sleep-promoting effects of _Sh_ require neither circadian clocks nor daily light cycles. By contrast, _Sh_ mutant phenotypes disappeared in LL at 29 °C, consistent with the lack of _Sh_ effects on L sleep in LD cycles at high temperature. Further sleep analyses revealed that _Sh_ mutants showed short DD sleep at either temperature (Fig. 2c, d) and no significant interactions between temperature and _Sh_ mutation were detected on DD sleep (P = 0.1002 for subjective L; P = 0.0918 for subjective D by Aligned ranks transformation ANOVA). These observations could be explained by a model that _Sh_ promotes sleep at either temperature but light masks _Sh_ effects more strongly at high temperature. In either constant condition, high temperature suppressed _Hk_ mutant sleep (Supplementary Fig. 3a, b) whereas ambient temperature did not significantly affect _sss_ mutant sleep (Supplementary Fig. 3c, d). Although sleep behaviors are likely governed by more complex mechanisms in LD cycles, our results demonstrate that light and temperature cooperatively gate the sleep-promoting effects of _Sh_ in a manner independent of circadian rhythms or sleep homeostasis. The interplay between light and temperature has also been documented on heat-induced sleep in circadian clock mutants[31].

**_Sh_ depletion in GABAergic neurons phenocopies _Sh_ mutants.** _Sh_ mutation leads to the hyperexcitability of affected neurons, delaying repolarization by potassium efflux and possibly elevating intracellular $Ca^{2+}$ levels[35,36]. Given the genetic evidence that _Sh_ promotes sleep, a simple hypothesis would be that loss of _Sh_ function in wake-promoting neurons excites their neural activity and thereby suppresses sleep. In fact, a previous study has shown

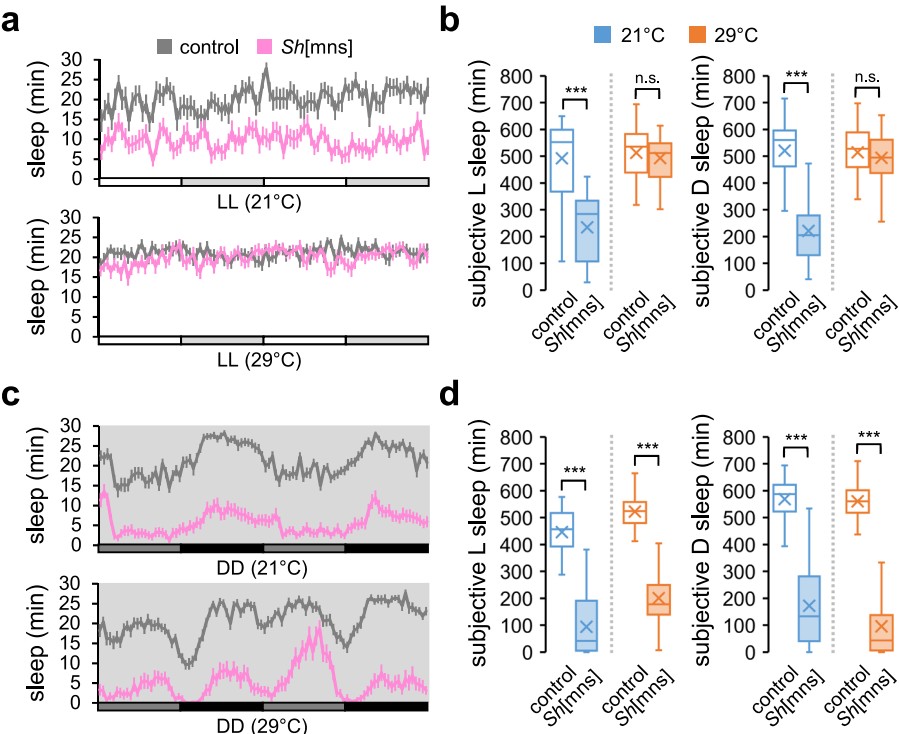

**Fig. 2 Light masks *Sh* effects on sleep at high temperature.** Sleep behaviors in individual male flies were analyzed in constant light (LL) or constant dark (DD) at 21 °C or 29 °C. **a**, **c** Sleep profiles of control (gray lines) or *Sh* mutants (pink lines) during the first two cycles of LL or DD. Data represent average ± SEM ($n = 20$–48). **b**, **d** Box plots represent the total amounts of subjective L or subjective D sleep on the second cycle of LL or DD at 21 °C (blue boxes) or 29 °C (orange boxes) ($n = 20$–48). Aligned ranks transformation ANOVA detected significant interactions between *Sh* mutation and temperature on sleep duration in LL ($P < 0.0001$ for either subjective L or subjective D), but not in DD ($P = 0.1002$ for subjective L; $P = 0.0918$ for subjective D). n.s., not significant; ***$P < 0.001$ as determined by Wilcoxon rank-sum test.

that RNA interference (RNAi)-mediated depletion of the SH complex, including HK and SSS, in sleep-promoting dFSB neurons suppresses sleep[21]. This puzzling observation indicates a more elaborate mechanism for *Sh* function in dFSB neurons that sustains their tonic firing during sleep[25]. Nonetheless, it does not exclude the possibility that *Sh* may also act in brain regions other than dFSB neurons[6], contributing to temperature-sensitive L sleep in *Sh* mutants. Accordingly, we employed a similar RNAi strategy to deplete *Sh* expression in a specific group of neurons and examined its effects on sleep behaviors.

We first confirmed that overexpression of the *Sh* RNAi transgene by the pan-neuronal galactose-responsive transcription factor 4 driver (ELAV-Gal4) reduced *Sh* transcript levels in adult fly heads (Supplementary Fig. 4). The pan-neuronal silencing of *Sh* expression indeed suppressed sleep only at 21 °C, partially phenocopying *Sh* mutant sleep (Fig. 3a, b). Similarly, pan-neuronal disruption of the *Sh* locus by CRISPR-mediated targeting[37] suppressed sleep at 21 °C, but not at 29 °C (Supplementary Fig. 5). The stronger transgenic phenotypes at low temperature cannot be explained by the intrinsic temperature dependency of the Gal4 activity since high temperature actually enhances the Gal4-driven expression of *Drosophila* transgenes in general. These results further support that loss of neuronal *Sh* function, but not the temperature-sensitive nature of the *Sh*[mns] allele, may underpin temperature-sensitive sleep phenotypes in *Sh* mutants.

We then tested a number of Gal4 drivers to deplete *Sh* expression in individual subsets of neurons (Fig. 3a and Supplementary Data 1). *Sh* depletion in wake-promoting neurons that release DA (TH-Gal4), OA (TDC2-Gal4), or acetylcholine (ChAT-Gal4)[38] negligibly affected sleep (Supplementary Fig. 6). We also did not detect any sleep suppression by *Sh* depletion in

dFSB neurons (23E10-Gal4) (Fig. 3a and Supplementary Fig. 7), possibly due to some differences in experimental conditions (e.g., temperature, gender, age). On the other hand, we found that two Gal4 drivers (i.e., 121y-Gal4 and 30y-Gal4) expressed broadly in adult brain, including the mushroom body, significantly suppressed L sleep at either temperature while causing short D sleep only at 21 °C (Supplementary Fig. 8). Accordingly, these data suggest that a sub-group of *Sh*-expressing neurons may suppress sleep regardless of ambient temperature. We also reason that sleep phenotypes observed in *Sh* mutants are caused by the net effects of its loss-of-function in various subtypes of *Sh*-expressing neurons (e.g., wake-promoting vs. sleep-promoting, constitutive vs. thermosensitive).

Our genetic screen further identified a vesicular γ-aminobutyric acid transporter (VGAT)-Gal4 driver as one of the strongest hits that mimicked L sleep phenotypes in *Sh* mutants (Fig. 3a, b). *Sh* depletion in VGAT-expressing neurons suppressed L sleep at 21 °C, but not at 29 °C, whereas it did not cause any D sleep phenotypes at either temperature. Similar results were obtained using independent transgenic lines of VGAT-Gal4 or *Sh* RNAi (Supplementary Fig. 9), excluding their off-target effects. The lack of D sleep phenotypes by VGAT-Gal4 may indicate that VGAT-expressing neurons contribute specifically to short L sleep in *Sh* mutants. Alternatively, the degree of *Sh* depletion by VGAT-Gal4 may not be sufficient to suppress D sleep. VGAT-Gal4 contains a transgenic promoter from the *VGAT* locus and is expressed in neurons that release the inhibitory neurotransmitter GABA[39]. To validate if *Sh* expression in GABAergic neurons was responsible for *Sh* mutant sleep, we genetically combined a glutamic acid decarboxylase 1 (GAD1)-Gal80 repressor transgene[40] with the pan-neuronal driver (ELAV-Gal4) or VGAT-Gal4 to overexpress the *Sh* RNAi transgene. Since GAD1 is a key enzyme for GABA

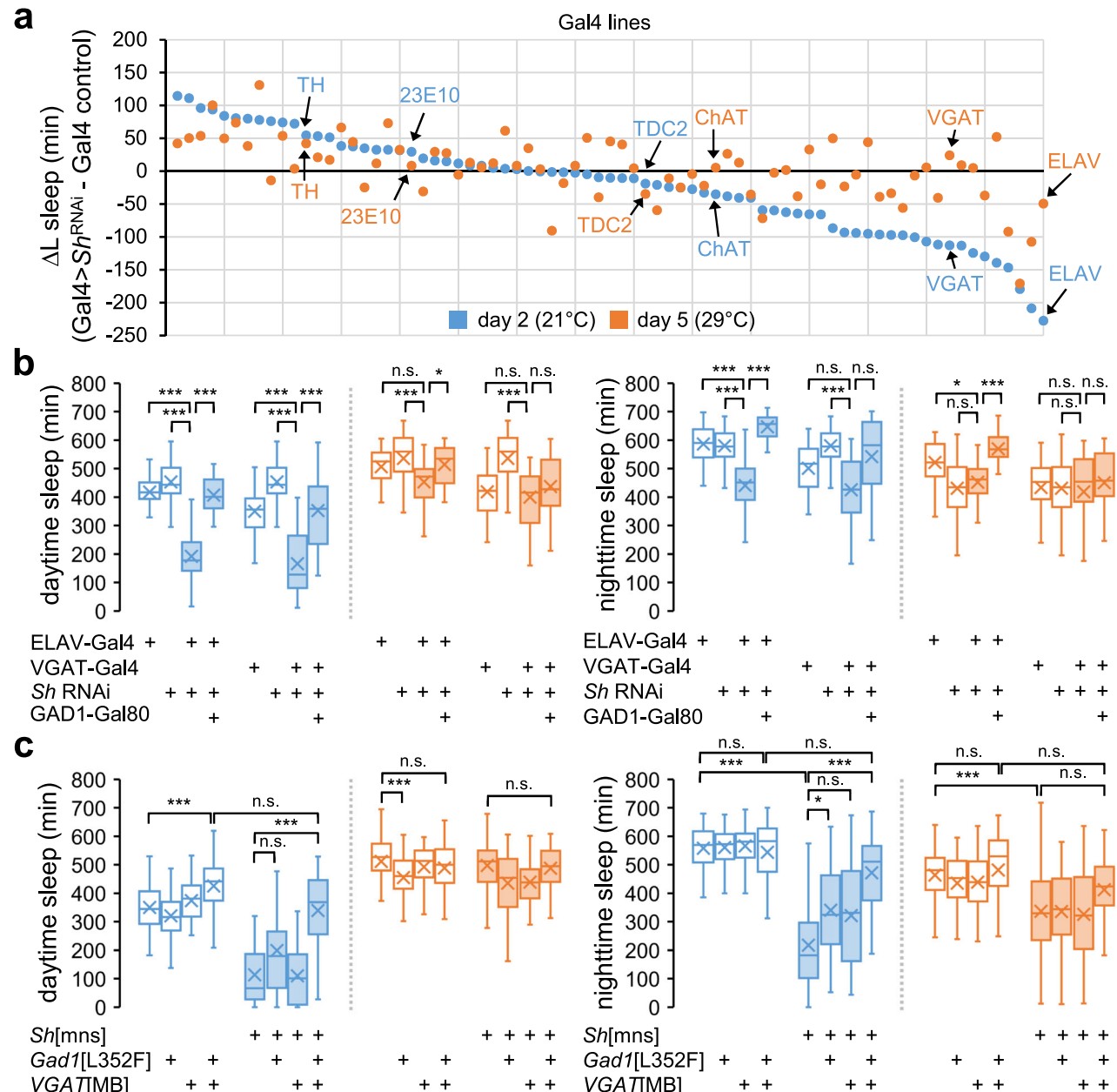

**Fig. 3 Sh acts in GABAergic neurons to promote sleep likely via GABA transmission. a** A series of Gal4 drivers was genetically combined with *Sh* RNAi transgene (*Sh* RNAi#1, BL53347) to deplete endogenous *Sh* expression in distinct sets of cells. Sleep behaviors were analyzed individually in transgenic male flies. The differences in the averaged amount of L sleep between Gal4>*Sh*^RNAi and Gal4 control flies (ΔL sleep, $n = 15$–91) for each Gal4 driver were plotted by blue (day 2, 21 °C) or orange dots (day 5, 29 °C), respectively. The x-axis indicates each Gal4 line sorted by the size of ΔL sleep at 21 °C. **b** Box plots represent the total amounts of L or D sleep on day 2 (21 °C, blue boxes) versus day 5 (29 °C, orange boxes) ($n = 24$–62). The *Sh* RNAi#1 transgene was co-expressed with *Dicer-2* to enhance RNAi effects. Aligned ranks transformation ANOVA detected significant interactions of temperature with *Sh* depletion in all neurons (ELAV-Gal4) or GABAergic neurons (VGAT-Gal4 (II), BL58980) on L sleep ($P < 0.0001$). n.s., not significant; *$P < 0.05$, ***$P < 0.001$ as determined by Wilcoxon signed-rank test. **c** Trans-heterozygous mutations in *GAD1* and *VGAT* suppress temperature-sensitive plasticity of wild-type and *Sh* mutant sleep ($n = 25$–122). n.s., not significant; *$P < 0.05$, ***$P < 0.001$ as determined by Aligned ranks transformation ANOVA, Wilcoxon signed-rank test.

synthesis, this genetic manipulation would block the RNAi-mediated depletion of *Sh* selectively in GABAergic neurons. Indeed, GAD1-Gal80 rescued the sleep phenotypes caused by the RNAi-mediated depletion of *Sh* (Fig. 3b). These data strongly implicate GABAergic neurons as a neural locus important for temperature-sensitive effects of *Sh* on sleep behaviors, although we do not rule out the possibility that VGAT-Gal4 or GAD1-Gal80 transgene may also be expressed in some non-GABAergic cells.

**Inhibition of GABA transmission suppresses *Sh* mutant sleep.** To validate *Sh*-dependent neural activity in GABAergic neuron, we expressed a Ca²⁺-sensitive transcriptional fluorescence reporter CaLexA[41] in VGAT-expressing neurons. Their fluorescence signals were then compared between wild-type or *Sh* mutant flies entrained at either 21 °C or 29 °C. Similar approaches have been employed in previous studies to measure long-term changes in intracellular Ca²⁺ levels as a proxy for neural

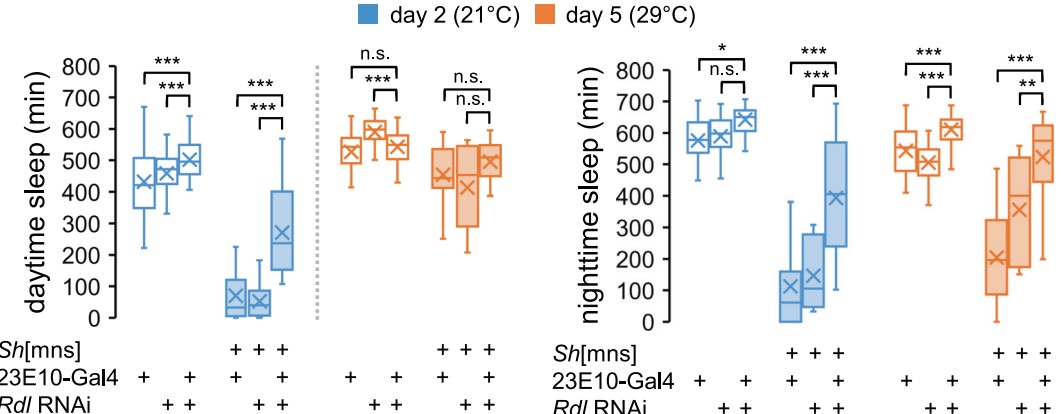

**Fig. 4 Ionotropic GABA receptor *Rdl* in dFSB neurons acts downstream of *Sh* to suppress sleep.** 23E10-Gal4 driver was genetically combined with *Rdl* RNAi transgene to deplete endogenous *Rdl* expression in dFSB neurons of wild-type or *Sh* mutant flies. Sleep behaviors were analyzed individually in transgenic male flies. Box plots represent the total amounts of L or D sleep on day 2 (21 °C, blue boxes) versus day 5 (29 °C, orange boxes) ($n = 20-63$). Aligned ranks transformation ANOVA detected significant interactions of *Rdl* depletion in dFSB neurons with *Sh* on sleep duration at 21 °C ($P < 0.001$ for L sleep; $P < 0.01$ for D sleep), but not at 29 °C. n.s., not significant; *$P < 0.05$, **$P < 0.01$, ***$P < 0.001$ as determined by Wilcoxon signed-rank test.

activity[22,42,43]. The quantification of our Ca$^{2+}$ imaging revealed that *Sh* mutation elevated intracellular Ca$^{2+}$ levels in subsets of VGAT-expressing neurons (Supplementary Fig. 10). This was consistent with previous observations that loss of *Sh* function increases neuronal excitability[35,36]. Significant effects of *Sh* on Ca$^{2+}$ levels were detected in mushroom body and ellipsoid body (Supplementary Fig. 10b, $P < 0.0001$ and $P = 0.0165$, respectively, by Aligned ranks transformation ANOVA). We further detected statistically significant interaction between *Sh* mutation and temperature on Ca$^{2+}$ levels in mushroom body ($P = 0.0148$ by Aligned ranks transformation ANOVA), but not in ellipsoid body ($P = 0.2180$ by Aligned ranks transformation ANOVA). In fact, high temperature generally weakened the CaLexA-driven fluorescence signals in either wild-type or *Sh* mutant brains (Supplementary Fig. 10b, $P < 0.0001$ for mushroom body and antenna lobe; $P < 0.001$ for ellipsoid body and pars intercerebralis by Aligned ranks transformation ANOVA). This observation possibly points to the intrinsic temperature-sensitivity of the CaLexA transgene. Nonetheless, these data support that *Sh* mutation likely increases the excitability of GABAergic neurons and thereby sustains GABA transmission.

If greater neural activity in GABAergic neurons was responsible for *Sh* mutant sleep, we reasoned that a down-scaling of the presynaptic GABA transmission should suppress *Sh* mutant phenotypes. To this end, we genetically silenced GABA transmission by heterozygous loss-of-function mutations in *VGAT*, which incorporates GABA into the synaptic vesicles of GABAergic neurons[39,44], or in GABA-synthesizing *GAD1*[45]. Their effects on sleep behaviors were then compared between wild-type and *Sh* mutant backgrounds. The heterozygosity of either *VGAT* or *GAD1* had negligible effects on wild-type sleep at either 21 °C or 29 °C (Fig. 3c). However, their trans-heterozygosity increased the amount of wild-type L sleep specifically at 21 °C (Fig. 3c, $P < 0.0001$ for temperature x genotype interaction on L sleep by Aligned ranks transforma-tion ANOVA), thereby dampening temperature-dependent changes in L sleep duration. More importantly, the trans-heterozygosity of *VGAT* and *GAD1* rescued *Sh* mutant sleep at 21 °C (Fig. 3c, $P < 0.0001$ for trans-heterozygosity of *VGAT* and *GAD1* × *Sh*[mns] interaction on either L or D sleep by Aligned ranks transformation ANOVA) while it had additive effects with *Sh* mutation on D sleep at 29 °C ($P > 0.05$ for trans-

heterozygosity of *VGAT* and *GAD1* × *Sh*[mns] interaction on D sleep by Aligned ranks transformation ANOVA). The hetero-zygosity of *VGAT* similarly suppressed L sleep phenotypes caused by *Sh* depletion in VGAT-expressing neurons (Supple-mentary Fig. 9b). Collectively, these genetic interaction data provide convincing evidence that a GABA-dependent mechan-ism contributes to temperature-dependent effects of *Sh* on sleep behaviors.

**Rdl in dFSB neurons acts downstream of Sh to suppress sleep.** GABA receptor agonists potently enhance sleep in both flies and mammals[11,12,46]. By contrast, we observed that genetic suppres-sion of GABA transmission actually promoted L sleep at 21 °C in wild-types and rescued short sleep phenotypes in *Sh* mutants. GABA receptor genes are broadly expressed in the adult fly brain[47,48]. Several lines of evidence indicate that the sleep-modulatory effects of each GABA receptor are mediated by a specific group of neurons involved in the regulation of sleep-wake cycles[12–16]. Given the inhibitory nature of GABA transmission onto postsynaptic neurons, we reasoned that GABA receptors may act in sleep-promoting neurons to mediate *Sh* mutant phe-notypes. To elucidate this sleep-suppressing role of GABA transmission, we depleted endogenous expression of individual GABA receptors in different groups of sleep-regulatory neurons and examined their effects on sleep behaviors.

The most striking effects were observed when we silenced the expression of the ionotropic GABA receptor, *Resistant to dieldrin* (*Rdl*), in sleep-promoting dFSB neurons[49]. While *Rdl* expression in dFSB neurons has been reported previously[50–52], we found that dFSB-specific *Rdl* depletion lengthened wild-type L sleep at 21 °C, but not at 29 °C (Fig. 4). These sleep phenotypes were in stark contrast to *Sh* mutant sleep (Fig. 1b, c), suggesting the opposing effects of presynaptic *Sh* in GABAergic neurons and postsynaptic *Rdl* in dFSB neurons on sleep behaviors. Indeed, *Rdl* depletion in dFSB neurons non-additively suppressed *Sh* mutant sleep only at 21 °C (Fig. 4, $P < 0.001$ or $P < 0.01$ for *Rdl* depletion × *Sh*[mns] interaction on L or D sleep, respectively, by Aligned ranks transformation ANOVA). These results support that *Rdl* in dFSB neurons acts genetically downstream of GABAergic *Sh* and antagonizes its effects on sleep at low temperature. Accordingly, *Sh* effects on sleep likely converge on a specific group of sleep-promoting neurons despite our mapping

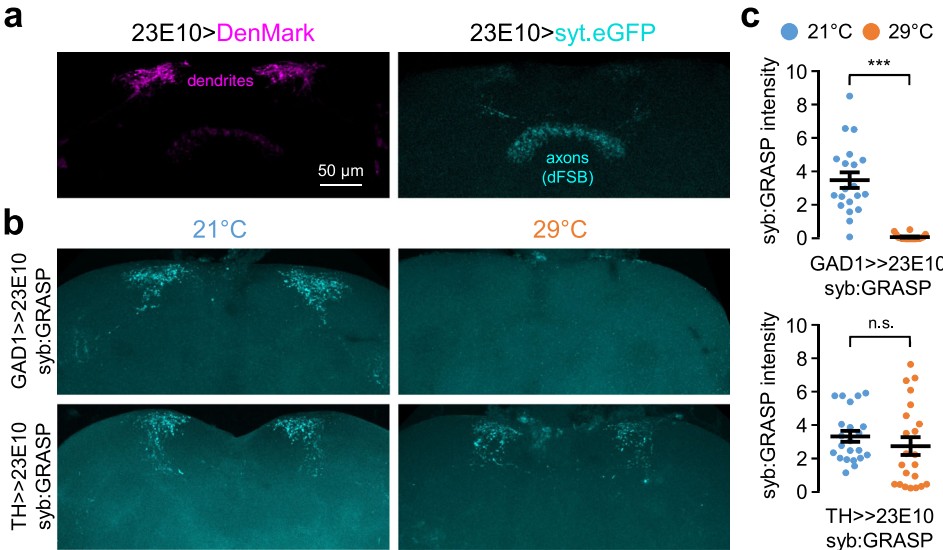

**Fig. 5 GABAergic synapses, but not dopaminergic synapses, onto dFSB neurons display temperature-sensitive activity. a** Dendrites and axonal projections from dFSB neurons were visualized by transgenic expression of the fluorescent marker proteins DenMark (magenta) and syt.eGFP (cyan), respectively (23E10-Gal4/UAS-DenMark, UAS-syt.eGFP). Weak DenMark signals in the dFSB region may indicate leaky expression in non-dendritic regions[80]. Alternatively, a minor population of dFSB neurons may have dendrites in the dFSB region. **b** Transgenic flies were pre-entrained in LD cycles at 21 °C (blue) or 29 °C (orange), and their whole brains were dissected out. The fluorescence images of whole-mount brains were obtained using a multi-photon microscopy. Representative z-stack images of the synaptic GRASP (syb:GRASP) signals from GABAergic (lexAop-nSyb-spGFP1-10, UAS-CD4-spGFP11/+; GAD1-LexA/23E10-Gal4) or dopaminergic neurons (lexAop-nSyb-spGFP1-10, UAS-CD4-spGFP11/+; TH-LexA/23E10-Gal4) to dFSB neurons were shown. **c** The fluorescence intensity of the syb:GRASP at the dendritic regions of dFSB neurons was quantified using ImageJ software. Fluorescence signals above a threshold level (i.e., background) were integrated from each hemisphere, and averaged per genotype at each temperature (n = 18–22 hemispheres). Data represent average ± SEM. n.s., not significant; ***P < 0.001 as determined by Mann–Whitney U test.

of the *Sh* RNAi phenotypes broadly to GABAergic neurons (Fig. 3b).

**GABAergic synapses onto dFSB display temperature-sensitivity**. To functionally validate GABAergic synapses onto dFSB neurons, we examined synaptic GFP reconstitution across synaptic partners (GRASP) between GAD1-expressing neurons and dFSB neurons. Unlike the original GRASP[53,54], synaptobrevin-GRASP (syb:GRASP) requires the fusion of synaptic vesicles with the presynaptic membrane to display trans-synaptic fluorescence at the synaptic cleft[55]. Accordingly, the presence of the syb:GRASP signals verifies the directionality of a given synapse between two groups of neurons. In addition, the relative intensity of their fluorescence is proportional to the neural activity of presynaptic neurons. With this experimental design, we detected the syb:GRASP signals enriched in the dendritic regions of dFSB neurons[56–59] (Fig. 5a, b), validating the directionality of functional synapses from GAD1-expressing neurons to dFSB neurons. Quantitative image analyses further revealed that the syb:GRASP signals from GAD1-expressing neurons were more evident in flies that had been entrained in LD cycles at 21 °C than in those entrained at 29 °C (Fig. 5c, P < 0.0001 by Mann–Whitney U test). These data suggest that GABAergic neurons display higher synaptic transmission onto dFSB neurons at low temperature.

Since dFSB neurons also receive inhibitory dopaminergic inputs[18,19,21], we examined whether ambient temperature also affected dopaminergic synapses onto dFSB neurons. In contrast to GABAergic synapses, the syb:GRASP signals from dopaminergic neurons were comparable between two groups of flies that had been entrained at either 21 °C or 29 °C, respectively (Fig. 5c, P = 0.1777 by Mann–Whitney U test). Taken together, our results indicate that high temperature specifically weakens GABAergic synapses onto dFSB neurons. We reason that this temperature-

dependent neural plasticity likely explains stronger sleep phenotypes in *Sh* mutants or *Sh*-depleted flies at low temperature.

**GABA and DA differentially reduce dFSB excitability**. DA transmission down-regulates the excitability of dFSB neurons, switches them to an electrically quiescent status, and suppress sleep[21]. On the other hand, our data indicate that GABAergic synapse onto dFSB neurons displays temperature-sensitive activity (Fig. 5) and supports temperature-dependent plasticity of wild-type and *Sh* mutant sleep (Figs. 3c and 4b). Given the inhibitory nature of GABA transmission to postsynaptic neurons, we asked if the excitability of dFSB neurons is differentially modulated by GABA and DA. To this end, we expressed an ATP-gated cation channel P2X_2 along with a synaptic calcium sensor sytGCaMP in dFSB neurons. The transgenic combination allowed us to cell-autonomously excite dFSB neurons by ATP application and measure their excitability using fluorescent $Ca^{2+}$ imaging in live brains[60]. We first found that ATP-induced elevation of intracellular $Ca^{2+}$ levels in dFSB neurons were comparable between transgenic flies entrained at 21 °C and 29 °C (Fig. 6). However, pre-incubation of either GABA or DA suppressed the excitability of dFSB neurons in a temperature-sensitive manner. In particular, GABA potently suppressed the excitability of dFSB neurons only at 21 °C (Fig. 6a) whereas DA silenced it only at 29 °C (Fig. 6b). These results indicate that inhibitory effects of GABA and DA on dFSB neurons are differentially gated by temperature. Considering our GRASP data, we reasoned that temperature might modify the neural property of postsynaptic dFSB neurons, thereby governing their excitability via temperature-specific neurotransmitter signaling.

**GABA transmission suppresses DA signaling in dFSB neurons**. To determine if ionotropic GABA transmission in postsynaptic dFSB neurons is affected by temperature, we examined GABA-dependent

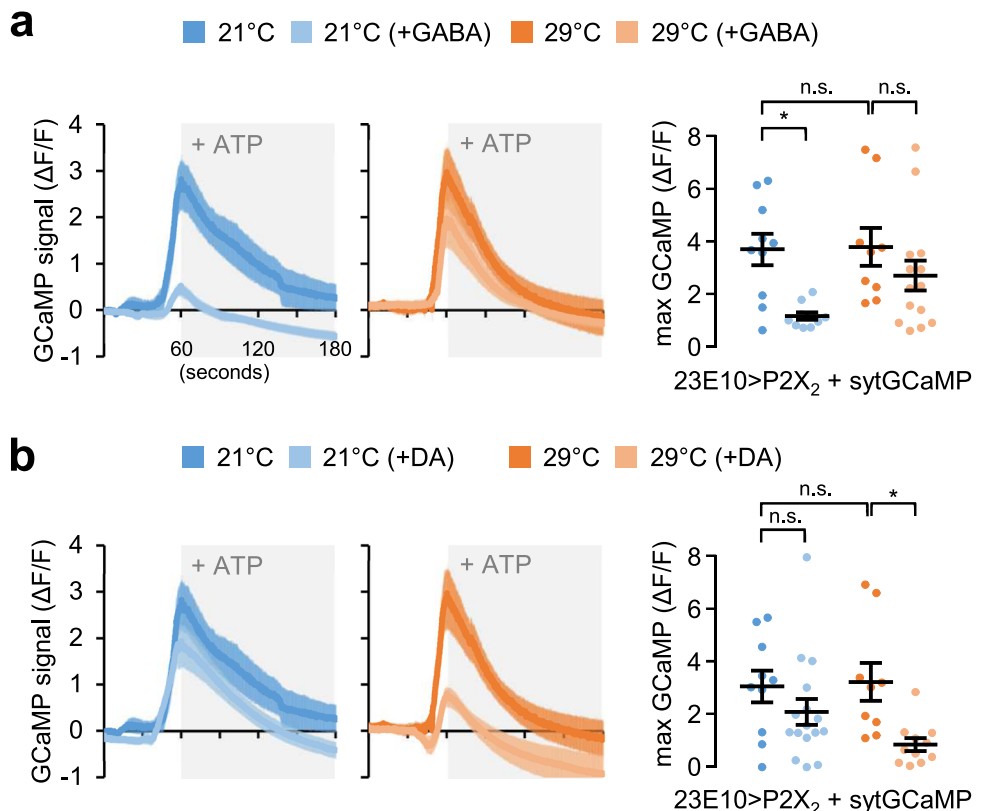

**Fig. 6 GABA and DA suppress the excitability of dFSB neurons in a temperature-sensitive manner.** An ATP-gated cation channel P2X₂ was expressed in dFSB neurons along with a synaptic calcium sensor sytGCaMP (23E10 > P2X₂, sytGCaMP) to quantify ATP-induced changes in intracellular $Ca^{2+}$ levels as an indirect readout of the neural excitability. Transgenic flies were pre-entrained in LD cycles at 21 °C (blue) or 29 °C (orange). Whole brains were dissected out, transferred to an imaging chamber, and equilibrated with HL3 buffer for 5 minutes. Where indicated, dissected brains were incubated with 50 mM GABA (**a**) or 10 mM DA (**b**) for 5 minutes prior to the batch application of 5 mM ATP (shaded by gray boxes). A time series of the fluorescence images was recorded using a photoactivated localization microscopy and their quantification was performed using ZEN software. Data represent average ± SEM ($n = 9$–14). n.s., not significant; *$P < 0.05$ as determined by Aligned ranks transformation ANOVA, Wilcoxon rank-sum test.

changes in intracellular chloride levels using a transgenic fluorescence resonance energy transfer (FRET) sensor SuperClomeleon[14,61]. Bath application of GABA robustly elevated chloride levels in dFSB neurons, but their FRET responses were comparable at 21 °C and 29 °C (Supplementary Fig. 11). It is thus unlikely that chloride influx via ionotropic GABA receptors is limiting at 29 °C to explain lack of GABA effects on the excitability of dFSB neurons at high temperature. We further quantified postsynaptic DA signaling in dFSB neurons using Epac1-camps, another transgenic FRET sensor for cyclic adenosine monophosphate (cAMP)[62] (Fig. 7a). cAMP is a signaling molecule downstream of D1-like DA receptors implicated in *Drosophila* sleep[18,19,21]. Live-brain imaging of the transgenic Epac1-camps, expressed in dFSB neurons, allowed us to measure the relative changes in intracellular cAMP levels by bath application of DA[19].

Interestingly, we observed that DA-induced cAMP elevation was more robust in dFSB neurons of transgenic flies entrained at 29 °C than those entrained at 21 °C (Fig. 7b). While these observations were consistent with temperature-sensitive effects of DA on the excitability of dFSB neurons, we reasoned that higher activity of GABAergic synapses at 21 °C might have a negative impact on DA signaling in dFSB neurons. Indeed, dFSB-specific depletion of *Rdl* reversed the repression of the intracellular cAMP response to DA at 21 °C, thus blunting its temperature-dependency (Fig. 7b). Moreover, pre-incubation with GABA repressed DA-induced cAMP elevation in dFSB neurons of control flies, while the transgenic depletion of *Rdl* masked GABA effects. The weaker suppression by exogenous GABA at 21 °C

likely reflected a floor effect caused by the stronger transmission of endogenous GABA to dFSB neurons at low temperature. Finally, we observed that an agonist of ionotropic GABA receptors (THIP), but not that of metabotropic GABA receptors (SKF-97541), suppressed DA-induced cAMP elevation (Fig. 7c), consistent with the *Rdl* RNAi effects. Collectively, our data demonstrate that ambient temperature tunes the postsynaptic signaling of DA transmission to dFSB neurons via temperature-sensitive GABA synapses. This neural mechanism may confer temperature-dependent plasticity on the sleep-regulatory function of dFSB neurons.

## Discussion

High ambient temperature promotes sleep during the daytime but suppresses sleep consolidation at night, thereby inducing greater nocturnal activity in diurnal animals[27,29]. The temperature-dependent remodeling of sleep-wake patterns represents a behavioral adaptation to the environment, which is considered beneficial for animal fitness and is likely conserved across species, including humans[63,64]. Nonetheless, the mechanisms underlying this phenomenon are poorly documented. Our genetic studies in *Drosophila* demonstrated that ionotropic GABA transmission from *Sh*-expressing neurons to sleep-promoting dFSB neurons constitutes as a neural pathway responsible for temperature-dependent control of sleep behaviors. The opposing effects of presynaptic *Sh* and postsynaptic GABA receptor *Rdl* may buffer drastic fluctuations in sleep duration over a range of ambient temperatures, thereby underscoring temperature-relevant sleep

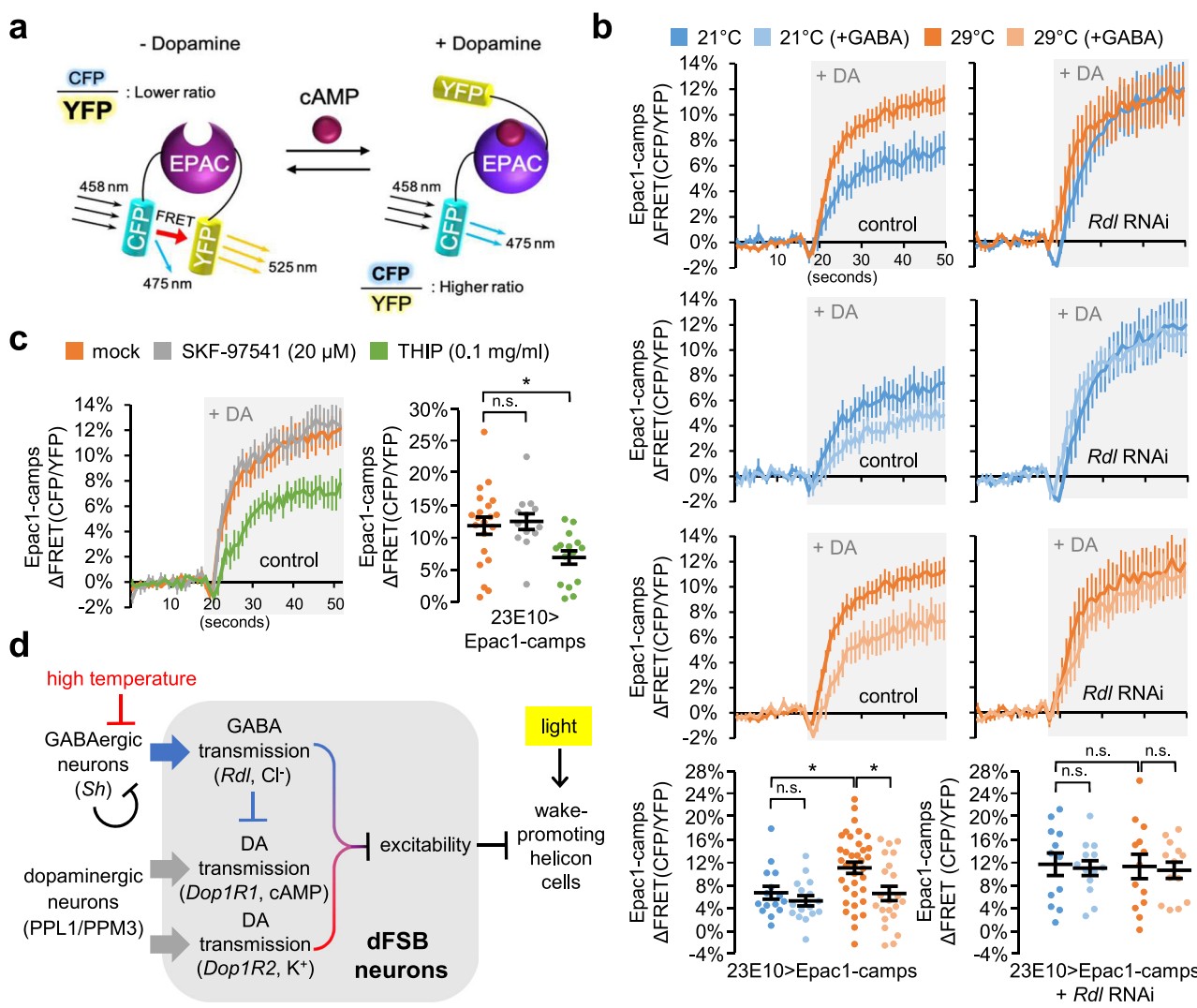

**Fig. 7 Ionotropic GABA transmission suppresses postsynaptic DA signaling in dFSB neurons and supports temperature-sensitive DA transmission.**
**a** DA receptor signaling elevates intracellular cyclic AMP (cAMP) levels. The subsequent binding of cAMP to Epac1-camps (EPAC) induces conformational change in the FRET sensor, thereby increasing the fluorescence ratio of CFP to YFP. **b** Epac1-camps was expressed in dFSB neurons of control (23E10 > Epac1-camps) or *Rdl* RNAi flies (23E10 > Epac1-camps + *Rdl* RNAi). Transgenic flies were pre-entrained in LD cycles at 21 °C (blue) or 29 °C (orange). Whole brains were dissected out and transferred to an imaging chamber. Where indicated, dissected brains were pre-incubated with 1 mM GABA for 5 min prior to the induction of FRET responses by the batch application of 10 mM DA (shaded by gray boxes). A time series of the fluorescence images was recorded using a multi-photon microscopy and their FRET analysis was performed using ZEN software. Data represent average ± SEM ($n$ = 15–37 for 23E10 > Epac1-camps; $n$ = 12–13 for 23E10 > Epac1-camps + *Rdl* RNAi). Two-way ANOVA detected significant effects of either temperature or GABA on the DA-induced FRET response in control ($P$ = 0.0238 or $P$ = 0.0151, respectively), but not in *Rdl*-depleted dFSB neurons ($P$ = 0.7894 or $P$ = 0.7040, respectively). n.s., not significant; *$P$ < 0.05 as determined by Tukey post hoc test. **c** 23E10 > Epac1-camps flies were pre-entrained in LD cycles at 29 °C. Where indicated, dissected brains were pre-incubated with SKF-97541 (20 µM, a metabotropic GABA receptor agonist) or THIP (0.1 mg/ml, an ionotropic GABA receptor agonist) for 5 min prior to the induction of FRET responses by the batch application of 10 mM DA. Data represent averag ± SEM ($n$ = 13–21). n.s., not significant; *$P$ < 0.05 as determined by one-way ANOVA, Tukey post hoc test. **d** A model for the genetic and neural interplay of sleep-promoting dFSB neurons that supports temperature- and light-sensitive plasticity of sleep behaviors.

homeostasis. In addition, the temperature-sensitive activity of the GABA synapse plays important roles in adjusting postsynaptic DA signaling in dFSB neurons, thereby modulating their neural property in a temperature-dependent manner. Finally, light-dependent masking overrides the sleep-regulatory output from this *Sh*-relevant pathway particularly at high temperature, explaining siestas on hot days. These findings uncover an important neural principle by which environmental changes are translated into synaptic plasticity of a specific neural locus with sleep-regulatory function, possibly leading to adaptive behavioral plasticity (Fig. 7d).

Previous studies have suggested that circadian clock genes (e.g., *period* and *cryptochrome*) and a group of circadian pacemaker neurons (e.g., posterior dorsal DN1p neurons) contribute to the immediate modification of sleep behaviors by elevated temperatures[30,31]. Our data indicate that temperature-adaptive sleep plasticity is manifested gradually in various sleep parameters after temperature shift (Supplementary Fig. 1). We reason that a chronic shift to high temperature implicates neural plasticity (e.-g., the adjustment of temperature-sensitive synapses) that stabilizes temperature-adaptive sleep behaviors. The transient effects of clock gene mutations on temperature-dependent sleep behaviors[31], as

well as the gender-specific delay in the latency to L sleep onset by high temperature[30], further support the idea that the molecular and neural mechanisms underlying the immediate behavioral responses to temperature shifts are likely distinguishable from those triggering long-term changes in sleep architecture.

A physiological range of environmental temperatures is sensed by dedicated groups of neurons that express distinct thermo-sensing molecules and are associated with specific behavioral responses (e.g., temperature preference or avoidance)[65]. This raises the possibility that *Sh*-expressing neurons are an intrinsic component for neural sensing or processing of ambient temperatures. Alternatively, but not exclusively, SH channel activity itself may be thermosensitive as observed in transient receptor potential (TRP) channels[66], thereby modulating its sleep-promoting effects in a temperature-dependent manner. However, SH activity seems to remain constant over a physiological range of temperatures, although genetic uncoupling of its voltage sensor from the gating mechanism can alter its temperature sensitivity[67]. In fact, we observed that a loss-of-function mutation in the thermosensitive channel *TrpA1* or the surgical ablation of antenna (a peripheral tissue for temperature sensing in *Drosophila*)[65] blunted the temperature effects on L sleep in wild-type flies (Supplementary Fig. 12, $P < 0.0001$ for temperature × *TrpA1* mutation interaction; $P < 0.001$ for temperature x antenna abla-tion interaction on L sleep by Aligned ranks transformation ANOVA). This was also consistent with the previously reported role of *TrpA1* in temperature-dependent control of morning wakefulness[30]. Nonetheless, *Sh* mutants displayed temperature-sensitive sleep phenotypes even in the absence of *TrpA1* (Sup-plementary Fig. 12, $P < 0.0001$ for temperature × *Sh* mutation interaction on either L or D sleep in *TrpA1* mutant backgrounds by Aligned ranks transformation ANOVA). These results suggest that *TrpA1*-dependent sensing of warm temperature may con-tribute to the behavioral plasticity of sleep in wild-types, whereas the *Sh*-relevant neural pathway likely acts independent or downstream of *TrpA1*.

dFSB neurons have been mapped as a postsynaptic locus that expresses *Drosophila* homologs of D1-like DA receptors and is responsible for DA-dependent arousal[18,19,21]. Genetic or phar-macological enhancement of DA transmission suppresses sleep via *Dop1R1*-mediated cAMP/PKA signaling in dFSB neurons[18,19]. Prolonged DA transmission switches dFSB neurons to electrically quiescent neurons[21]. This transformation is transduced by *Dop1R2* and involves neural manipulation of two types of potassium currents manifested by the voltage-gated *Sh* and the leak channel *Sandman*, respectively[21]. Nonetheless, we identified presynaptic GABAergic neurons expressing *Sh* and postsynaptic dFSB neurons expressing *Rdl* as the thermosensitive neural pathway important for gating the postsynaptic DA transmission in dFSB neurons as well as reorganizing sleep behaviors according to ambient temperature.

Previous studies in mammals have actually made several observations relevant to our findings. The inhibitory synapse from a group of wake-promoting GABAergic neurons in lateral hypothalamus to the VLPO, one of the mammalian sleep-promoting nuclei analogous to dFSB neurons, likely acts as a sleep-wake switch[68]. Cross-talks between GABA and DA trans-mission have also been demonstrated at the level of postsynaptic receptor signaling[69]. For example, ionotropic GABA receptors directly associate with hippocampal D5 receptors; this suppresses DA-induced cAMP levels and GABA currents[70]. In addition, D1 receptor activation leads to PKA-dependent phosphorylation of ionotropic GABA receptors and their subsequent internaliza-tion[71–73]. Given the latter observation, we speculate that GABAergic inhibition of DA receptor signaling in dFSB neurons provides a neural mechanism ensuring robust switching between these two postsynaptic pathways upon temperature shifts. More specifically, at 21 °C, GABA may strengthen its own transmission by inhibiting DA receptor signaling. Weaker GABA transmission at 29 °C would reverse the repression of DA transmission, which in turn would suppress any residual GABA receptor activity. This model explains temperature-sensitive dominance of GABA or DA in silencing the excitability of dFSB neurons (Fig. 6). Nonetheless, we found that chloride influx induced by exogenous GABA was not limiting in dFSB neurons at 29 °C (Supplementary Fig. 11), suggesting the presence of additional postsynaptic mechanisms that might suppress GABA effects on the dFSB excitability at high temperature.

dFSB neurons suppress wake-promoting helicon cells via the neuropeptide allatostatin-A signaling[24]. This neural pathway constitutes an auto-regulatory circuit that comprises dFSB neu-rons, helicon cells, and R2 ring neurons of the ellipsoid body (R2 EB)[22,23,57] and establishes a neural basis of sleep homeostasis in *Drosophila*[3,74,75]. Intriguingly, light excites helicon cells[24] and thus we reason that low baseline activity of helicon cells in the absence of light may cause a floor effect, desensitizing them to the inhibitory signals from dFSB neurons (Fig. 7d). Under these conditions, *Sh* mutant sleep would be contributed by other neural pathways. The presence of light, on the other hand, may increase the baseline activity of helicon cells that can be further scaled by opposing effects of the inhibitory signals from dFSB neurons, and of temperature-sensitive GABAergic synapse onto dFSB neurons. This model explains light-dependent masking of *Sh* mutant phenotypes at 29 °C since high temperature will silence wake-promoting effects of *Sh* mutation in GABAergic neurons on dFSB neurons. Future studies should further elaborate on principles underlying the complex interaction of temperature with different light regime on *Sh* mutant sleep.

In conclusion, plasticity in sleep behaviors represents an important strategy for sustaining animal physiology in response to external or internal changes in the sleep environment. Our study establishes a genetic pathway that constitutes temperature-sensitive GABA transmission to the sleep-promoting neural locus and generates neural plasticity underpinning the adaptive organization of sleep architecture. Given the similarities of temperature-dependent sleep behaviors between flies and humans, we propose that this underlying neural strategy may be conserved across species.

## Methods

**Fly stocks**. All flies were maintained in standard cornmeal–yeast–agar medium at 25 °C. w[1118] (BL5905; wild-type control), *Sh*[mns] (BL24149), *Sh*[TKO] (BL76968), *sss*[P1] (BL16588), *sss*[MIC] (BL34309), *Gad1*[L352F] (BL6295), *VGAT* [MB01219] (BL23022), *TrpA1*[1] (BL26504), UAS-*Sh* RNAi #1 (BL53347), UAS-*Rdl* RNAi (BL52903), ELAV-Gal4 (BL458), ELAV-Gal4; UAS-Cas9 (BL67073), VGAT-Gal4 (II) (BL58980), VGAT-Gal4 (III) (BL58409), 23E10-Gal4 (BL49032), TH-Gal4 (BL8848), TDC2-Gal4 (BL9313), ChAT-Gal4 (BL6798), 121y-Gal4 (BL30815), 30y-Gal4 (BL30818), GAD1-LexA (BL60324), UAS-mLexA-VP16-NFAT (BL66542; CaLexA), UAS-DenMark, UAS-syt.eGFP (BL33065), lexAop-nSyb-spGFP1-10, UAS-CD4-spGFP11 (BL64315; syb:GRASP), UAS-SuperClomeleon (BL59847), and UAS-Epac1-camps (BL25407) were obtained from Bloomington *Drosophila* Stock Center. *Hk*[1] (101-119) was obtained from Kyoto Stock Center. UAS-*Sh* RNAi #2 (v104474) was obtained from Vienna *Drosophila* Resource Center. GAD1-Gal80, TH-LexA, and UAS-sytGCaMP6s; UAS-P2X₂ were gifts from Y. Li (Chinese Academy of Sciences), H. Tanimoto (Tohoku University), and R. Allada (Northwestern University), respectively.

**Behavioral analysis**. To avoid possible effects of undefined genetic backgrounds on sleep behaviors, all mutants and transgenic flies were tested in outcrossed conditions. Sleep behaviors in experimental genotypes were compared to those in control genotypes obtained from appropriate genetic crosses in parallel. w[1118] was set as a wild-type in all genetic crosses to generate control flies heterozygous for mutant alleles or transgenes (e.g., Gal4, Gal80, UAS). Hemizygous male mutants (e.g., *Sh*[mns], *Hk*[1]) were generated by crossing mutant virgins to w [1118] males whereas their genetic controls were obtained from parallel crosses in reverse orientation. Sleep behaviors in experimental flies with a specific

combination of mutant alleles or transgenes were compared to those in control flies heterozygous for individual ones. Each male fly was transferred into a 65 × 5 mm glass tube containing 5% sucrose and 2% agar food. Locomotor activity in individual flies was then recorded using the *Drosophila* Activity Monitor system (Trikinetics) and quantified by the number of infrared beam crosses per minute. Behavioral data were collected from more than two independent tests and averaged per genotype. For LD sleep analysis, flies were entrained for 3 days in LD cycle at 21 °C prior to the temperature shift to 29 °C at ZT16 (lights-on at ZT0; lights-off at ZT12) in the last LD cycle at 21 °C. They were further incubated for 3 days in LD cycle at 29 °C. For LL or DD sleep analysis, flies were kept in constant conditions at 21 °C or 29 °C. A sleep bout was defined as a behavioral episode during which flies did not show any activity for 5 min or longer. Sleep parameters were accordingly analyzed with an Excel macro[76].

**Whole-mount brain imaging**. Transgenic flies were entrained in LD cycles at either 21 °C or 29 °C for a week. Whole brains were dissected in phosphate-buffered saline (PBS), fixed in PBS containing 3.7% formaldehyde for 28 minutes at room temperature (RT), and then washed three times in PBS containing 0.3% Triton X-100 (PBS-T). For immunostaining, fixed brains were blocked in PBS-T containing 0.5% normal goat serum for 30 min at RT and then incubated with mouse anti-GFP antibody (diluted in PBS-T containing 0.5% NGS and 0.05% sodium azide at 1:1000, NeuroMab) for 2 days at 4 °C. After washing with PBS-T, brains were further incubated with anti-mouse Alexa Flour 488 antibody (diluted at 1:600, Jackson ImmunoResearch) for 1 day at 4 °C, washed with PBS-T, and then mounted in a VECTASHIELD mounting medium (Vector Laboratories). Fluorescence images were acquired using a multi-photon microscope (LSM780NLO, Carl Zeiss) and analyzed using ImageJ software.

**Live-brain imaging**. Transgenic flies pre-entrained in LD cycles at either 21 °C or 29 °C for a week were anesthetized in ice. A whole brain was briefly dissected in hemolymph-like HL3 solution (5 mM HEPES pH 7.2, 70 mM NaCl, 5 mM KCl, 1.5 mM CaCl$_2$, 20 mM MgCl$_2$, 10 mM NaHCO$_3$, 5 mM trehalose, 115 mM sucrose) and then placed on a cover glass in a magnetic imaging chamber (Chamlide CMB, Live Cell Instrument) filled with HL3 buffer. Live-brain imaging was recorded at room temperature. For GCaMP-based calcium imaging, brains were equilibrated with HL3 buffer for 5 min and then incubated with 50 mM GABA or 10 mM dopamine for 5 min prior to the batch application of 5 mM ATP. A time series of the fluorescence images was recorded using a photoactivated localization microscopy (ELYRA P.1, Carl Zeiss) with a C-Apochromat 40×/1.20 W Korr M27 at a pixel resolution of 512 × 512. Calcium signals were quantified by background substitution method and analyzed using ZEN software (Carl Zeiss). For SuperClomeleon-based FRET imaging, brains were equilibrated with HL3 buffer for 10 minutes prior to the induction of FRET responses by 50 mM GABA. A time series of the fluorescence images was recorded using a multi-photon microscope (LSM780NLO, Carl Zeiss) with a Plan-Apochromat 20×/0.8 at a pixel resolution of 512 × 512. The power of a 458 nm-laser projection was 10%. Two filter ranges (473–491 nm and 509–535 nm) were set for ECFP and EYFP channels, respectively. Pinhole was set as 13.69 AU to select the region of interest within the axon bundle of dFSB neurons. For Epac1-camps-based FRET imaging, brains were incubated with GABA (1 mM), THIP (4,5,6,7-tetrahydroisoxazolo(5,4-c)pyridin-3-ol, 0.1 mg/ml), or SKF-97541 (20 μM) for 5 min prior to the induction of FRET responses by 10 mM DA. A time series of the fluorescence images was recorded using a multi-photon microscope (LSM780NLO, Carl Zeiss) with a Plan-Apochromat 40×/1.3 oil lens at a pixel resolution of 256 × 256. Each stack contained five slices with ~10 μm of step sizes. Pinhole was fully opened (19.07 AU) to avoid any subtle z-drift during imaging acquirement. FRET image analyses were performed using ZEN software (Carl Zeiss).

**Quantitative transcript analysis**. Fifty fly heads per genotype were homogenized in TRIzol Reagent and total RNAs were extracted according to the manufacturer's instructions (Thermo Fisher Scientific). After DNase I digestion, purified RNAs were reverse-transcribed using M-MLV Reverse Transcriptase (Promega) and random hexamers. Quantitative real-time PCR was performed using 2x Prime Q-Mastermix (GeNet Bio) with the diluted cDNA samples and gene-specific sets of primers in LightCycler 480 real-time PCR system (Roche). The primer sequences used in this study were as follows: 5'-CCG GTC AAT GTC CCT TTA GA-3' (forward) and 5'-CTC GAA GAG CAG CCA GAC TT-3' (reverse) for *Sh*; 5'-ATC TCC CAC AGG ACG TCA AC-3' (forward) and 5'-GCG ACG AAG AGA AGG ATC AC-3' (reverse) for *pabp* (internal control).

**Statistics and reproducibility**. All statistical analyses were performed using GraphPad Prism 6 or R (version 3.5.3). Datasets that did not pass either Shapiro-Wilk test for normality ($P < 0.05$) or Brown-Forsythe test for equality of variances ($P < 0.05$) were analyzed by Aligned ranks transformation ANOVA with ARTool library for multifactorial statistical analyses[77,78]. Aligned rank transformation (ART) allowed the assessment of temperature or genotype effects on sleep behaviors using non-parametric datasets to validate their significant interaction by ANOVA. Unless otherwise indicated, all sleep parameters were analyzed by repeated measures one-way or two-way ANOVA. For sleep analyses in LL or DD,

ordinary two-way ANOVA was conducted. Interaction comparisons were performed using contrast function from the *emmeans* package (α = 0.05 with the Bonferroni correction)[79]. Post hoc multiple comparisons were performed by Wilcoxon signed-rank test for repeated measures or by Wilcoxon rank-sum test for unpaired conditions (α = 0.05 with the Bonferroni correction). For statistical analyses of imaging data, significant differences or interactions between experimental conditions were determined by Student's *t*-test, one-way ANOVA, or two-way ANOVA. Where normality or homoscedasticity test was not passed, Mann–Whitney *U* test or Aligned ranks transformation ANOVA was conducted accordingly. The numbers of samples analyzed per condition and *P* values obtained from individual statistical analyses were indicated in the main text or figure legends.

**Reporting summary**. Further information on research design is available in the Nature Research Reporting Summary linked to this article.

## Data availability

The data that support the findings of this study are included in the paper or available from the corresponding author upon request.

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

## Acknowledgements

We thank A. Guo, Y. Li, H. Tanimoto, R. Allada, Bloomington *Drosophila* Stock Center, Korea *Drosophila* Resource Center, Kyoto Stock Center, and Vienna *Drosophila* Resource Center for fly strains. This work was supported by grants from the Suh Kyungbae Foundation (SUHF-17020101); from the National Research Foundation (NRF) funded by the Ministry of Science and ICT (MSIT), the Republic of Korea (NRF-2017R1E1A2A02066965; NRF-2018R1A5A1024261; NRF- 2018H1A2A1063084).

## Author contributions

C.L. conceived the study; J.K., Y.K., and C.L. designed experiments; J.K., Y.K., M.S.H., and C.L. generated transgenic flies and performed sleep analyses; Y.K., H.L., and J.-H.H. performed imaging experiments; J.K. and Y.K. performed transcript analyses; J.K., B.B., and D.N., and C.L. performed statistical analyses; J.K., Y.K., and C.L. wrote the manuscript.

## Competing interests

The authors declare no competing interests.

## Additional information

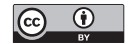

