## [Peer Review File · Communications Biology]

Reviewers' comments:

Reviewer #1 (Remarks to the Author):

This manuscript investigates the genetic and neural mechanisms of sleep modulation by temperature. The authors start with an interesting observation that sleep reduction in Sh mutants is temperature and light dependent. They then show Rdl knockdown in sleep-promoting dorsal fan-shaped body (dFSB) partially rescues Sh sleep effects, and GABAergic synapses onto dFSB neurons exhibit temperature-dependent activity. These findings suggest Sh functions in GABAergic, dFSB-projecting neurons to modulate sleep in a temperature-dependent manner. In addition, they find that GABA signaling suppresses dopamine-dependent cAMP elevation in dFSB neurons. They propose a model in which switching between dopamine and GABA signaling in dFSB neurons underlies temperature-dependent changes in sleep architecture. These results are novel and interesting and the model is intriguing but they raise a few questions. Specific questions and suggestions are listed below.

1. It is unclear what the effect of GABA on R23E10 excitability is. Since GABA is an inhibitory neurotransmitter and Rdl knockdown in dFSB leads to a small but significant increase in nighttime sleep (Fig. 4b), it appears GABA inhibits excitability of dFSB neurons. However, if GABA suppresses DA-induced inhibition of dFSB neurons, it would imply that the net effect of GABA would be to increase dFSB neuron excitability. Additional experiments measuring neuronal excitability in response to GABA and/or DA would address the question. At a minimum, the question should be discussed.
2. The interaction between light and temperature on Sh effects is complex and needs further explanation. Sh mutants show reduced sleep at high and low temperatures in constant darkness but only at a low temperature in LD (in both L and D periods) and LL. The model (Fig. 6D) places the light input downstream of Sh-expressing neurons that mediate temperature dependent GABA signaling. How this model explains the observed light and temperature effects should be discussed in more detail.
3. The authors suggest that "a sub-group of Sh-expressing neurons may promote sleep regardless of ambient temperature" (page 5). If so, why don't Sh mutants, which presumably have lost of Sh function in those neurons, lose sleep at 29C?
4. VGAT-Gal4 may not be completely specific for GABAergic neurons. Simultaneously knocking down Sh and GAD1 could confirm that Sh acts in GABAergic neurons to mediate temperature-dependent modulation of sleep.
5. The authors suggest that Sh knockdown in dFSB neurons may have had little effects in their hands because the RNAi line they used may not be as efficient as the one used previously by Donlea et al. For this reason, and because it is generally a good practice to use multiple RNAi lines to address potential off-target effects, it would be important to repeat key experiments using the Sh RNAi line used by Donlea et al.
6. Fig. 3a does not label most of the Gal4 lines. A supplemental table listing the Gal4s and statistical tests would be helpful.
7. Fig. 4a should show dendritic and axonal markers, not just assert that different processes represent axons and dendrites.
8. Fig. 1b, why were sss heterozygotes used as controls? sss Df/+ seems different from sss P1/+, suggesting Df/+ may contain haplo-insufficient deletions. Control vs sssP1 homozygotes would be better.
9. Fig. 2, what happens to Hk and sss mutants in LL and DD?
10. For cAMP measurement. what was the temperature during recording?

Reviewer #2 (Remarks to the Author):

This study by Kim et al examine the role an important potassium channel, Shaker (Sh) in regulating membrane properties and thus activity of the neuronal circuit modulating sleep levels under different ambient temperatures based on presence or absence of light. They propose the role of GABA in inhibiting wakefulness by inhibiting the activity of dopaminergic dorsal Fan Shaped Body neurons.

They provide nuanced information regarding the control of sleep, which has previously been mostly restricted to standard light: dark conditions at 25 deg C. The studies involve the use of whole body mutants followed by tissue specific knockdown by RNAi for the initial findings for the role of Sh. They also use two reporters of neuronal activity namely – CalexA and Epac1 to test the hypotheses that a certain set of neurons are active specifically under warm temperature, and to establish dopamine induced activity of dFSB. Overall, the study is informative and of interest to the field.

I have a few broad and comments. General comments:

The authors have provided details of the statistical methods used and the graphs are also very detailed, which is nice, however, considering this is a fly sleep study its important to know how many replicate experiments were performed, if at all and if this is based on only one experiment each.

Since ANOVAs have been performed on the data, it would be expected that the authors tested for the following: a) independence of cases- however, I suspect that it cannot be met, because the data from the same flies are being compared for 21 and 29 deg C; (b) normal distribution of data; (c) equal variance across groups (homoscedasticity). Please verify and clarify. Based on figure legends there appears to be unequal sample sizes also. These violations can be dealt with by consulting someone with expertise in biostatistics.

The authors also do not provide information regarding the genetic backgrounds of their various mutant alleles and also the transgenic lines used for tissue specific gene expression. Since sleep levels are notoriously affected by genetic background, it is essential that the authors indicate how they ensured that backgrounds are largely similar when testing across genotypes.

Specific comments:

Introduction, pg 2 last para: “Modifications of the sleep architecture in mmediate response to heat implicate circadian clock-dependent mechanism”. This is not clear at all, how, in fact the 2 sentences together would suggest a non-clock dependent temperature sensitive response, which need not invoke involvement of the circadian clock at all. Please rewrite.

Pg 3: Its not clear what the authors mean by

“behavior in a single session”, data is collected for longer than 2 days.

“behavioural plasticity” is also vague

Pg3 para2, L12: “L sleep” reference is figure S1b?

Pg3 para2, L13: “deconsolidated D sleep” which figure? What is the evidence for deconsolidation?

Pg3 para2, L14: nocturnal activities “correlated with” loose use of the term and its not clear from the references what the authors wish to convey.

Pg3 para2, L15: “ambient temperature did not have an effect on locomotion” cannot be claimed, only activity counts are estimated, we cannot conclude from this that locomotion per se is unaffected.

Supp S2: the n ranges from 19 to 122 is that an error? If not have they corrected for unbalanced design in the ANOVA?

Pg3 para3: “short D sleep partially rescued by high temperature” is not supported by the test shown. Same line, suggestion: replace ‘whereas’ with “and”

Pg3 para3: “to our surprise” since we just saw that Shaker mutants have a different phenotype at warm 29 deg C, it's no longer surprising. Suggest change.

Pg 6 para 1: ‘..non-additively rescued..’ it is not clear what additive or non-additive means in this context and also in several other instances that follow.

“..had additive effects on D sleep..P=0.1071” it's not clear also what the authors are intending to convey.

Pg 6 2nd para The authors have screened for different GABA-R and different sleep circuit driver lines, but data is not shown? Last para: ‘modestly lengthened is a value of about 150 mins, while in a later comparison “... non-additively suppressed...” also have a difference of about 150 mins, is this not a modest suppression?

Pg 7: The syb GRASP genotypes must be clearly indicated. The result, method, legend all fail to convey how in this study one can infer ‘dendritic regions’. The original paper by McPhearson clearly gives the genotype which allows one to make this inference.

Pg 7 3rd para: “...on the other hand..” please provide figure references here.

Figure 5 legend: the top row must be described before the middle row. Genotype information is insufficient.

Please find our point-by-point responses below in blue.

[Editorial comments]

Your manuscript entitled "The voltage-gated potassium channel *Shaker* promotes sleep via thermosensitive GABA transmission" has now been seen by 2 referees. You will see from their comments below that while they find your work of interest, some important points are raised. We are interested in the possibility of publishing your study in *Communications Biology*, but would like to consider your response to these concerns in the form of a revised manuscript before we make a final decision on publication.

We therefore invite you to revise and resubmit your manuscript, taking into account the points raised. Please highlight all changes in the manuscript text file.

Specifically, we ask that you provide additional experiments to show neuronal excitability in response to GABA, and that you test for off-target effects in your RNAi lines, as requested by Reviewer #1. We also ask that you address Reviewer #2's points regarding missing tests in your ANOVA analyses.

[Reviewers' comments]

Reviewer #1 (Remarks to the Author):

This manuscript investigates the genetic and neural mechanisms of sleep modulation by temperature. The authors start with an interesting observation that sleep reduction in *Sh* mutants is temperature and light dependent. They then show *Rdl* knockdown in sleep-promoting dorsal fan-shaped body (dFSB) partially rescues *Sh* sleep effects, and GABAergic synapses onto dFSB neurons exhibit temperature-dependent activity. These findings suggest *Sh* functions in GABAergic, dFSB-projecting neurons to modulate sleep in a temperature-dependent manner. In addition, they find that GABA signaling suppresses dopamine-dependent cAMP elevation in dFSB neurons. They propose a model in which switching between dopamine and GABA signaling in dFSB neurons underlies temperature-dependent changes in sleep architecture. These results are novel and interesting and the model is intriguing but they raise a few questions. Specific questions and suggestions are listed below.

1. It is unclear what the effect of GABA on R23E10 excitability is. Since GABA is an inhibitory neurotransmitter and *Rdl* knockdown in dFSB leads to a small but significant increase in nighttime sleep (Fig. 4b), it appears GABA inhibits excitability of dFSB neurons. However, if GABA suppresses DA-induced inhibition of dFSB neurons, it would imply that the net effect of GABA would be to increase dFSB neuron excitability. Additional experiments measuring neuronal excitability in response to GABA and/or DA would address the question. At a minimum, the question should be discussed.

To examine GABA or DA effects on the excitability of dFSB neurons, we expressed an ATP-gated cation channel P2X2 along with a synaptic calcium indicator *syt-GCaMP* in dFSB neurons. This allowed us to cell-autonomously excite dFSB neurons by ATP application and

measure their excitability using fluorescent Ca²⁺ imaging in live brains (Yao et al. 2012 J Neurophysiol 108:684). We first found that ATP-induced elevation of intracellular Ca²⁺ levels in dFSB neurons were comparable between transgenic flies entrained at 21°C and 29°C (Fig. 6). However, pre-incubation of either GABA or DA suppressed the excitability of dFSB neurons in a temperature-sensitive manner. In particular, GABA suppressed the excitability of dFSB neurons more evidently at 21°C (Fig. 6a) whereas DA silenced it only at 29°C (Fig. 6b). These pieces of new evidence suggest that inhibitory effects of GABA and DA on dFSB neurons are differentially gated by temperature. We discussed our new findings along with other imaging data to support our conclusion in the revised manuscript.

2. The interaction between light and temperature on *Sh* effects is complex and needs further explanation. *Sh* mutants show reduced sleep at high and low temperatures in constant darkness but only at a low temperature in LD (in both L and D periods) and LL. The model (Fig. 6D) places the light input downstream of *Sh*-expressing neurons that mediate temperature dependent GABA signaling. How this model explains the observed light and temperature effects should be discussed in more detail.

We revised our text in Discussion to better explain *Sh* mutant sleep based on our model.

3. The authors suggest that "a sub-group of *Sh*-expressing neurons may promote sleep regardless of ambient temperature" (page 5). If so, why don't *Sh* mutants, which presumably have lost of *Sh* function in those neurons, lose sleep at 29C?

We reason that sleep phenotypes observed in *Sh* mutants are caused by the net effects of its loss-of-function in various sub-types of *Sh*-expressing neurons (e.g., wake-promoting vs. sleep-promoting, constitutive vs. thermosensitive). Accordingly, *Sh* mutant phenotypes should not necessarily be identical with those caused by *Sh* depletion in individual groups of *Sh*-expressing neurons (Fig. 3a). We modified our text to avoid any confusion in the revised manuscript.

4. VGAT-Gal4 may not be completely specific for GABAergic neurons. Simultaneously knocking down *Sh* and *GAD1* could confirm that *Sh* acts in GABAergic neurons to mediate temperature-dependent modulation of sleep.

Unfortunately, we found that RNAi-mediated depletion of *GAD1* or *VGAT* using *VGAT-Gal4* caused lethality in our experimental conditions. Nonetheless, these observations may be supportive of the *VGAT-Gal4* expression in GABAergic neurons given the essential function of GABA transmission. We instead employed an independent *VGAT-Gal4* line to confirm that *Sh* depletion by *VGAT-Gal4* causes short L sleep in a temperature-dependent manner (Supplementary Fig. 9b). We further showed that the heterozygosity of *VGAT* mutation (Supplementary Fig. 9b) or *GAD1-Gal80* transgene (Fig. 3b) suppressed sleep phenotypes in *VGAT>Sh* RNAi flies. Since these pieces of evidence were not definitive, we modified our text to discuss the reviewer concern above.

5. The authors suggest that *Sh* knockdown in dFSB neurons may have had little effects in their hands because the RNAi line they used may not be as efficient as the one used previously by Donlea et al. For this reason, and because it is generally a good practice to use multiple RNAi lines to address potential off-target effects, it would be important to repeat key experiments using the *Sh* RNAi line used by Donlea et al.

The *Sh* RNAi line used by Donlea et al. (*Sh* RNAi #2, v104474) encodes a 137-bp target region that covers a 21-bp target sequence of our original *Sh* RNAi line (*Sh* RNAi #1, BL53347). Further analyses of transcript levels and sleep behaviors confirmed that both RNAi lines pan-neuronally depleted *Sh* expression (Supplementary Fig. 4) and caused temperature-dependent short L sleep in combination with VGAT-Gal4 (Supplementary Fig. 8a), although *Sh* RNAi #1 gave somewhat stronger phenotypes. Nonetheless, we failed to detect any sleep phenotypes by dFSB-specific depletion of *Sh* using either RNAi line (Supplementary Fig. 7). We reasoned that differences in experimental conditions (e.g., temperature, gender, age) might explain these inconsistent observations and revised our manuscript accordingly.

6. Fig. 3a does not label most of the Gal4 lines. A supplemental table listing the Gal4s and statistical tests would be helpful.

We included Supplementary Table 1 in the revised manuscript that listed all Gal4 lines in Fig. 3a and showed significance values obtained from our statistical analyses of individual Gal4 data.

7. Fig. 4a should show dendritic and axonal markers, not just assert that different processes represent axons and dendrites.

The neural anatomy of dFSB neurons has been well established by previous studies (Donlea et al. 2018 *Neuron* 97:378, Qian et al. 2017 *Elife* 6:e26519). Nonetheless, we included our own data in the revised manuscript that visualized dendritic and axonal regions in dFSB neurons by the transgenic expression of DenMark and *syt:eGFP*, respectively (Fig. 5a, top).

8. Fig. 1b, why were *sss* heterozygotes used as controls? *sss Df/+* seems different from *sss P1/+*, suggesting *Df/+* may contain haplo-insufficient deletions. Control vs *sssP1* homozygotes would be better.

Since we had difficulties in obtaining *sss* homozygotes even in outcrossed genetic backgrounds, we generated trans-heterozygous *sss* mutants and compared their sleep behaviors to those in heterozygous controls. To address the reviewer's concern regarding *sss Df* backgrounds, we employed two insertional alleles (i.e., *sss[P1]* and *sss[MI]*) that displayed comparable sleep behaviors in heterozygous conditions. Their trans-heterozygous mutants showed short L sleep in LD cycles at 29°C (Fig. 1b and c) and LL did not rescue their L sleep at either temperature (Supplementary Fig. 3c and d). Accordingly, *sss* mutant sleep was distinguishable from *Sh* or *Hk* mutant sleep, consistent with our original findings using *sss/Df* trans-heterozygotes.

9. Fig. 2, what happens to *Hk* and *sss* mutants in LL and DD?

Hk mutant sleep was observed only at 21°C in constant conditions whereas *sss* mutant sleep was detectable in a temperature-insensitive manner (Supplementary Fig. 3).

10. For cAMP measurement. what was the temperature during recording?

All live-brain imaging was recorded at room temperature. We revised our text in Methods accordingly.

Reviewer #2 (Remarks to the Author):

This study by Kim et al examine the role an important potassium channel, Shaker (Sh) in regulating membrane properties and thus activity of the neuronal circuit modulating sleep levels under different ambient temperatures based on presence or absence of light. They propose the role of GABA in inhibiting wakefulness by inhibiting the activity of dopaminergic dorsal Fan Shaped Body neurons. They provide nuanced information regarding the control of sleep, which has previously been mostly restricted to standard light: dark conditions at 25 deg C. The studies involve the use of whole body mutants followed by tissue specific knockdown by RNAi for the initial findings for the role of Sh. They also use two reporters of neuronal activity namely – *CalexA* and *Epac1* to test the hypotheses that a certain set of neurons are active specifically under warm temperature, and to establish dopamine induced activity of dFSB. Overall, the study is informative and of interest to the field. I have a few broad and comments.

General comments:

1. The authors have provided details of the statistical methods used and the graphs are also very detailed, which is nice, however, considering this is a fly sleep study its important to know how many replicate experiments were performed, if at all and if this is based on only one experiment each.

We collected our data from independent experiments--typically three or more but some data in Gal4 screen (Fig. 3a) were averaged from two independent behavioral tests. We revised our text in Methods accordingly.

2. Since ANOVAs have been performed on the data, it would be expected that the authors tested for the following: a) independence of cases- however, I suspect that it cannot be met, because the data from the same flies are being compared for 21 and 29 deg C; (b) normal distribution of data; (c) equal variance across groups (homoscedasticity). Please verify and clarify. Based on figure legends there appears to be unequal sample sizes also. These violations can be dealt with by consulting someone with expertise in biostatistics.

In our revised manuscript, we performed Shapiro-Wilk test and Levene's test to check normality and equal variance across groups, respectively. Any datasets that did not pass these tests were analyzed by Aligned ranks transformation ANOVA with ARTool library for multifactorial statistical analyses. This method could also be applied to datasets obtained from repeated measures using an error term in the formula argument that ARTool provides (Yap et al. 2017 Nat Commun 8, 1815; Cooper et al. 2018 Curr Biol 28, 2940). More detailed description of our statistical analyses was included in Methods and figure legends.

3. The authors also do not provide information regarding the genetic backgrounds of their various mutant alleles and also the transgenic lines used for tissue specific gene expression. Since sleep levels are notoriously affected by genetic background, it is essential that the authors indicate how they ensured that backgrounds are largely similar when testing across genotypes.

Sleep behaviors are affected by genetic backgrounds and the iso-genization of genetic backgrounds by repeatedly outcrossing to a wild-type background will be one option to avoid this issue. We actually employed quite a few isogenized Gal4 lines in our *Sh* RNAi mapping experiments and found that different Gal4/+ heterozygous controls still display substantial variations in their sleep amounts. In fact, many notorious effects of genetic backgrounds disappear substantially by single outcrosses unless they are dominant. We thus tested all mutants and transgenic flies in outcrossed conditions, and compared their behaviors to those obtained from appropriately paired, control genetic crosses in parallel. We also validated the original phenotypes using independent genetic strategies or resources (e.g., chromosomal deficiency, independent allele, independent transgene) where applicable. Some of those data, however, were not included in the current manuscript for conciseness. We revised our method to better clarify our genetic strategy.

Specific comments:

4. Introduction, pg 2 last para: “Modifications of the sleep architecture in immediate response to heat implicate circadian clock-dependent mechanism”. This is not clear at all, how, in fact the 2 sentences together would suggest a non-clock dependent temperature sensitive response, which need not invoke involvement of the circadian clock at all. Please rewrite.

We revised our text to clarify this.

5. Pg 3: Its not clear what the authors mean by “behaviors in a single session”, data is collected for longer than 2 days. “behavioural plasticity” is also vague

To clarify this, we deleted “in a single session” in the revised manuscript. We do not feel “behavioral plasticity” is vague since 1) temperature-dependent behavioral plasticity is defined in the last paragraph on page 2; and 2) the second paragraph on page 3 also describes “behavioral plasticity” in more detail.

6. Pg3 para2, L12: “L sleep” reference is figure S1b? Pg3 para2, L13: “deconsolidated D sleep” which figure? What is the evidence for deconsolidation? Pg3 para2, L14: nocturnal activities “correlated with” loose use of the term and its not clear from the references what the authors wish to convey.

We revised our text and figure references in the second paragraph on page 3 to improve its readability and clarity.

7. Pg3 para2, L15: “ambient temperature did not have an effect on locomotion” cannot be claimed, only activity counts are estimated, we cannot conclude from this that locomotion per se is unaffected.

We actually examined waking activity (Supplementary Fig. 1g) since this sleep parameter is considered as a quantitative proxy for general locomotion in *Drosophila* sleep studies. Nonetheless, we toned down the original text in the revised manuscript to address the reviewer concern above.

8. Supp S2: the n ranges from 19 to 122 is that an error? If not have they corrected for unbalanced design in the ANOVA?

Regarding our statistical analyses, please see our response to the reviewer #2, comment #2.

9. Pg3 para3: “short D sleep partially rescued by high temperature” is not supported by the test shown. Same line, suggestion: replace ‘whereas’ with “and”

We revised the text and employed an appropriate statistical method to support our original conclusion.

10. Pg3 para3: “ to our surprise” since we just saw that Shaker mutants have a different phenotype at warm29 deg C, its no longer surprising. Suggest change.

We removed it from the original text.

11. Pg 6 para 1: ‘..non-additvely rescued..’ it is not clear what additive or nonadditive means in this context and also in several other instances that follow. “..had additive effects on D sleep..P=0.1071” its not clear also what the authors are intending to convey.

Our definition of additive or non-additive effects is based on statistically non-significant or statistically significant interactions, respectively, between two components--this could be between two different genetic backgrounds or between genotype and temperature. We believe this terminology is generally accepted in the field but we revised our text where necessary.

12. Pg 6 2nd para The authors have screened for different GABA-R and different sleep circuit driver lines, but data is not shown? Last para: ‘ modestly lengthened is a value of about 150 mins, while in a later comparison “... nonadditively suppressed...” also have a difference of about 150 mins, is this not a modest suppression?

Our initial tests were conducted only in a wild-type background and some combinations of the Gal4/GABA-R RNAi transgenes caused lethality, excluding them from behavioral analyses. Accordingly, we did not build up this dataset in a publishable quality but we included those data crucial for our follow-up directions in the manuscript. “non-additively” described any statistically significant interaction between two components (in this case, we validated statistically different effects of dFSB-specific *Rdl* depletion on sleep amount in control versus *Sh* mutants), not the size of their difference (please see our response to the reviewer #2, comment #11). In this sense, “non-additively” was not a description of our data comparable to “modestly”. Nonetheless, we removed all “modestly” from our manuscript to avoid any unnecessary confusion.

13. Pg 7: The syb GRASP genotypes must be clearly indicated. The result, method, legend all fail to convey how in this study one can infer ‘dendritic regions’. The original paper by McPhearson clearly gives the genotype which allows one to make this inference.

We indicated the syb:GRASP genotypes in the figure legend of our revised manuscript. The neural anatomy of dFSB neurons has been well established by previous studies (Donlea et al. 2018 *Neuron* 97:378, Qian et al. 2017 *Elife* 6:e26519). Nonetheless, we included our own data in the revised manuscript that visualized dendritic and axonal regions in dFSB neurons by the transgenic expression of DenMark and syt:eGFP, respectively, using 23E10-Gal4 driver.

14. Pg 7 3rd para: “ ...on the other hand..” please provide figure references here. Figure 5 legend: the top row must be described before the middle row. Genotype information is insufficient.

We added figure references to the paragraph, and reorganized the figure legend, and provided genotype information as the reviewer suggested.

REVIEWERS' COMMENTS:

Reviewer #2 (Remarks to the Author):

The authors addressed all of my previous concerns. I have only a few minor comments.

In Fig. 5, it appears there is some DenMark signal in the dFSB. Please show the red and green channels separately, and discuss the implications of R23E10 dendrites in the dFSB. Also, the upper right image in Fig. 5 is grainy.

Fig. 4a seems unnecessary.

For bar graphs with small N's (e.g., GCaMP and FRET data), it would be better to show individual data in a column scatter format.

Reviewer #3 (Remarks to the Author):

The authors have addressed most of my major concerns.

Please find our point-by-point responses below in blue.

[Reviewers' comments]

Reviewer #2 (Remarks to the Author):

The authors addressed all of my previous concerns. I have only a few minor comments.

1. In Fig. 5, it appears there is some DenMark signal in the dFSB. Please show the red and green channels separately, and discuss the implications of R23E10 dendrites in the dFSB. Also, the upper right image in Fig. 5 is grainy.

We showed two channel images for DenMark and syt.eGFP signals, respectively, and removed the upper right/grainy image from the revised manuscript to better organize panels in Fig. 5. In addition, references for the R23E10 dendrites were appropriately cited in the text while we briefly discussed the implications of R23E10 dendrites in the dFSB region in the figure legend.

2. Fig. 4a seems unnecessary.

We removed Fig. 4a from the revised manuscript.

3. For bar graphs with small N's (e.g., GCaMP and FRET data), it would be better to show individual data in a column scatter format.

We converted all bar graphs to the dot-plot format in the revised manuscript.

Reviewer #3 (Remarks to the Author):

The authors have addressed most of my major concerns.